# Near-critical spreading of droplets

Raphael Saiseau [1,2] ✉, Christian Pedersen [3], Anwar Benjana[1], Andreas Carlson [3], Ulysse Delabre[1], Thomas Salez [1] ✉ & Jean-Pierre Delville [1] ✉

We study the spreading of droplets in a near-critical phase-separated liquid mixture, using a combination of experiments, lubrication theory and finite-element numerical simulations. The classical Tanner's law describing the spreading of viscous droplets is robustly verified when the critical temperature is neared. Furthermore, the microscopic cut-off length scale emerging in this law is obtained as a single free parameter for each given temperature. In total-wetting conditions, this length is interpreted as the thickness of the thin precursor film present ahead of the apparent contact line. The collapse of the different evolutions onto a single Tanner-like master curve demonstrates the universality of viscous spreading before entering in the fluctuation-dominated regime. Finally, our results reveal a counter-intuitive and sharp thinning of the precursor film when approaching the critical temperature, which is attributed to the vanishing spreading parameter at the critical point.

The spreading of viscous droplets on solid substrates has been extensively studied over the last decades[1–3]. For droplet sizes smaller than the capillary length[2], the viscocapillary regime yields self-similar asymptotic dynamics, i.e. the so-called Tanner's law[4], with the droplet radius $R$ increasing in time $t$ as $-t^{1/10}$. To establish this scaling, two ingredients are invoked: global volume conservation, and a local balance at the contact line between driving capillary forces and viscous dissipation in the liquid wedge. However, such a continuous description implies a finite change of the fluid velocity over a vanishing height at the contact line, and thus leads to an unphysical divergence of viscous stress and dissipation[5]. To solve this paradox, a microscopic molecular-like cut-off length is required, and appears through a logarithmic factor in Tanner's law. In this spirit, theoretical and experimental investigations introduced various possible regularization mechanisms[1,6–14], including a gravito-capillary transition[15–18], surface roughness[17], thermal effects[19], Marangoni-driven flows[20], diffusion[21,22], or a slip condition at the solid substrate[23]. In the particular case of total wetting, the existence of a thin precursor film ahead of the contact line has been proposed as the main candidate[24–26]. However, despite tremendous efforts to measure the microscopic length, or to characterize the associated logarithmic factor, the problem is still open. Conversely, solving the free-interface dynamical evolution of a droplet-like perturbation on a thin liquid film in the lubrication approximation

showed that Tanner's law can be considered as a negligible-film-thickness limit of capillary levelling[27]. Such a statement was further comforted by its extension to the gravity-driven[28] and elastic-bending-driven[29] cases. As such, it is possible to unambiguously determine the microscopic precursor-film thickness from the spreading of any droplet in total-wetting and lubrication conditions.

In this article, we investigate droplet spreading in a near-critical phase-separated binary liquid[30], with four main objectives. First and most importantly, close to a fluid-fluid critical point, an isotropic liquid belongs to the {$d = 3$, $n = 1$} universality class of the Ising model, where $d$ and $n$ are, respectively, the space and order-parameter dimensions, so that the results are immediately generalizable to any fluid belonging to the same universality class. Secondly, critical phenomena are often accompanied by a wetting transition at a temperature which might be either identical or distinct from the critical one[3], so precursor films can also be investigated near the critical point. Thirdly, as many fluid properties vary with the proximity to a critical point according to power-law variations of the type $\sim (\Delta T/T_c)^\alpha$, with $\Delta T = T - T_c$ the temperature distance to the critical point $T_c$, and $\alpha$ some positive or negative critical exponent, the spreading dynamics may be continuously and precisely tuned by varying the temperature. Finally, our study also provides evidence for droplet spreading in a liquid environment, which was scarcely studied[31].

[1]Univ. Bordeaux, CNRS, LOMA, UMR 5798, Talence F-33400, France. [2]Laboratoire Matière et Systèmes Complexes, UMR 7057, CNRS, Université Paris Cité, Paris F-75006, France. [3]Mechanics Division, Department of Mathematics, University of Oslo, Oslo 0316, Norway. ✉e-mail: raphael.saiseau@gmail.com; thomas.salez@cnrs.fr; jean-pierre.delville@u-bordeaux.fr

# Results

The experimental configuration is depicted in Fig. 1. We use a water-in-oil micellar phase of microemulsion[32,33], as described in detail in the Supplementary Method 1 section. Briefly, at the chosen critical composition (water, 9% wt, toluene, 79% wt, SDS, 4% wt, butanol, 17% wt), it exhibits a low critical point at $T_c$ close to 38 °C, above which the mixture separates into two phases of different micelle concentrations, $\Phi_1$ and $\Phi_2$ with $\Phi_2 < \Phi_1$ (see Fig. 1a). The microemulsion is enclosed in a tightly-closed fused-quartz cell of 2 mm thickness (Hellma 111-QS.10X2) which is introduced in a home-made thermally-controlled brass oven with four side-by-side windows. As working in a tight cell is mandatory with critical fluids, we use a contactless optical method to create a wetting drop at the bottom wall. Note that the microemulsion is transparent (absorption coefficient smaller than $4.6 \times 10^{-4}$ cm$^{-1}$) at the employed wavelength, which prevents any laser-induced heating effect. The sample is set at a temperature $T > T_c$ and a continuous frequency-doubled Nd$^{3+}$ − YAG (wavelength in vacuum $\lambda = 532$ nm, TEM$_{00}$ mode) laser beam is focused on the meniscus of the phase-separated mixture using a ×10 Olympus® microscope objective (NA = 0.25). The photon momentum mismatch between the two phases, proportional to the refraction-index contrast, generates a radiation pressure, and the interface thus bends (see Fig. 1b) as a result of the balance between the latter with hydrostatic and Laplace pressures[34]. As the interfacial tension $\gamma$ of near-critical interfaces vanishes at the critical point, with $\gamma = \gamma_0 (\Delta T / T_c)^{2\nu}$, where $\nu = 0.63$ and $\gamma_0 = 5.0 \times 10^{-5}$ N/m in our case, the interfacial deformation can be made very large. When the beam propagates downwards and with sufficient power, the interface can become unstable (see Fig. 1c) due to the total reflection of light within the deformation. In this case, a jet is formed, with droplets emitted at the tip[35,36]. Note that the jetting power threshold can also be used to measure the interfacial tension[35]. The length of the jet can be tuned with the laser power to bring its tip close to the bottom of the cell, without touching it. Then, by reducing the power, the jet breaks up into many droplets due to the Rayleigh-Plateau instability. By increasing again the power, below the jetting threshold, the laser beam forces coalescence between several droplets to produce a large one which can be further pushed by radiation pressure towards a borosilicate substrate placed at the bottom of the cell (see Fig. 1d). We turn off the laser just before contact, and follow the droplet spreading using ×20 or ×50 Olympus® microscope objectives, with resolutions of 1.0 and 0.8 μm respectively, and a Phantom® VEO340L camera for the frame grabbing. Note the existence of a prewetting film on the substrate, at least up to $\Delta T = 15$ K[37].

Figure 1d displays two image sequences corresponding to the coalescence and spreading of droplets at $\Delta T = 8$ and 1 K. The spreading time scale comparatively increases by approximately one order of magnitude for $\Delta T = 1$ K, as a result of the vanishing interfacial tension near $T_c$. We also notice that the droplet volumes reduce over time, indicating the presence of evaporation, as expected for finite-size objects in an environment at thermodynamic equilibrium. We stress here that we employ the standard terminology "evaporation" all along the article, despite the outer fluid is not a "vapor" but another liquid-like system. At early stages, both profiles display large curvature gradients. Since our focus here is on the long-term asymptotic spreading behavior, we define the temporal origin $t = 0$ from a first experimental image where the curvature is homogeneous, except near the contact-line region, and a spherical-cap fit is valid.

Each image sequence is then treated using a custom-made auto-matized contour detection based on a Canny-threshold algorithm, where the droplet profiles correspond to the external maxima of the intensity gradients (see Supplementary Method 2). Spherical-cap fits allow to extract the droplet volume $V(t)$, radius $R(t)$, and apparent contact angle $\theta(t)$, which are then averaged using a custom-made exponentially-increasing time window to get a logarithmically-distributed data set (see Supplementary Method 2). In the inset of Fig. 2a, we plot the experimental droplet volume as a function of time, for four different values of $\Delta T$. In all cases, the volume decreases until the droplet is fully evaporated. By using the initial volumes $V_0$ and by extrapolating the times $t_f$ of final evaporation for all droplets, we then plot in the main panel the same data in dimensionless form, with $V/V_0$ as a function of $1 - t/t_f$. We observe a data collapse onto a unique power-law behavior, with fitted exponent 1.77, which is close to the 11/7 value theoretically predicted for evaporating droplets[12]. In Fig. 2b, we further plot the contact radius $R$, normalized by its initial value $R_0$, as a function of time, for all $\Delta T$. A Tanner-like power law systematically emerges at intermediate times, until evaporation eventually dominates the evolution.

To model the observed spreading dynamics, we consider a large initial droplet-like interfacial perturbation profile $d(r, t = 0)$, with $r$ the radial coordinate, atop a flat thin-film of thickness $\epsilon$, and describe its evolution through the profile $d(r, t)$ at all times, in the small-slope limit within the lubrication approximation[2]. Therein, a horizontal Newtonian viscous flow of viscosity $\eta$ is driven by the gradient of Laplace pressure. Since most of the dissipation occurs in the wedge-like region near the apparent contact line[5], we further make an approximation and neglect the influence of viscous shear stresses in the surrounding

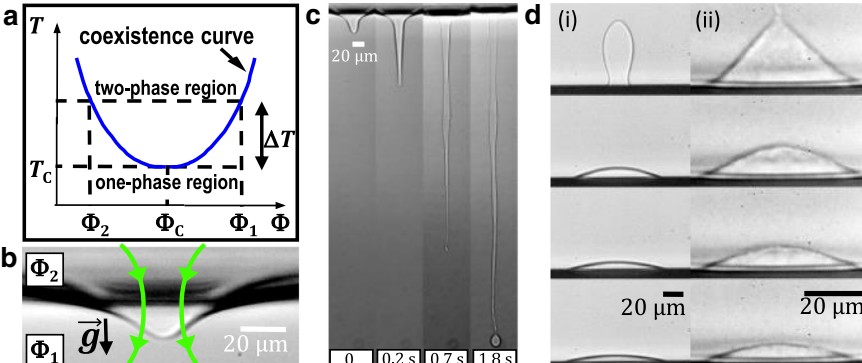

**Fig. 1 | Experimental system. a** Schematic phase diagram of the used binary liquid mixture (i.e., a micellar phase of the microemulsion, see Supplementary Method 1), where $T$ is the temperature, $\Phi$ the micelle concentration, and $T_c$ and $\Phi_c$ the coordinates of the critical point. **b** Radiation pressure-induced optical bending of the interface separating the two coexisting phases at $T > T_c$, where the downward laser beam is represented by the arrows. **c** Image sequence of the optical jetting instability with drop formation at the tip. **d** Image sequences of a less-dense-phase droplet of concentration $\Phi_2$ coalescing and spreading over a borosilicate substrate placed at the bottom of the cell, when surrounded by the denser phase of concentration $\Phi_1$. The temperature distances to the critical point $T_c$, the initial droplet volumes, and the time intervals between images are: (i) $\Delta T = 8$ K, $V_{ini} = 30.3$ pL, $dt = 3$ s; (ii) $\Delta T = 1$ K, $V_{ini} = 21.5$ pL, $dt = 20$ s.

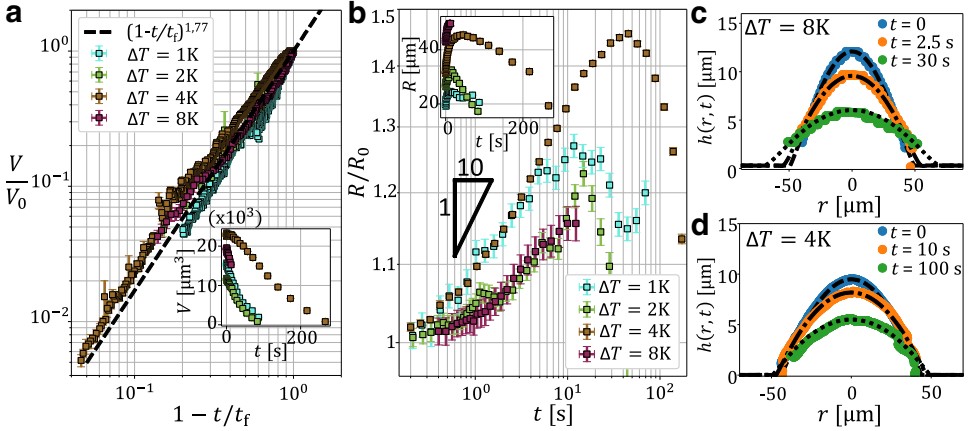

**Fig. 2 | Raw data: profiles and main observables. a** Rescaled droplet volume $V/V_0$ as a function of the rescaled time $1 - t/t_f$, with $V_0$ the initial volumes and $t_f$ the evaporation times of all droplets, for four different distances to the critical temperature $\Delta T$. The dashed line indicates the empirical power law $(1 - t/t_f)^{1.77}$. Inset: corresponding raw data. **b** Contact radius, divided by its initial value $R_0$, as a function of time for the same temperatures. The 1/10 power-law exponent of Tanner's law is indicated with a slope triangle. Inset: corresponding raw data. Error bars on droplet volume are derived from the errors on droplet height and contact radius, which are described in Supplementary Method 2. **c, d** Droplet profiles at different times obtained from experiments (symbols) and compared to the numerical solutions of Eq. (1) (dashed/dotted lines) for $\Delta T = 8$ K (**c**) and $\Delta T = 4$ K (**d**). Source data are provided as a Source Data file.

phase. Note that this hypothesis would not hold if the atmospheric fluid was much more viscous than the droplet fluid[31]. Fortunately, this is not the case near the critical point, where both viscosities are almost equal. The evolution is then described by the axisymmetric capillary-driven thin-film equation[27]:

$$\partial_t h + \frac{\gamma}{3\eta r}\partial_r\left[rh^3\partial_r\left(\frac{1}{r}\partial_r h + \partial_r^2 h\right)\right] = \mathcal{H}(1 - R)f,\qquad(1)$$

where $h(r, t) = \epsilon + d(r, t)$ is the free-interface height from the solid substrate, $\mathcal{H}(1 - R)$ is the Heaviside step function with $R(t)$ the advancing radius of the droplet, and $f(t)$ is an added coefficient accounting for evaporation. The latter is chosen in order to precisely mimic the experimentally-measured evaporation of the droplet (see Fig. 2a). Note that the Heaviside function ensures that the prewetting film—which is at thermodynamical equilibrium in contrast to the droplet—does not evaporate. Equation (1) is numerically integrated using a finite-element solver[38]. The experimental radial profiles depicted in Fig. 1d are chosen as initial profiles, after angular averaging and smoothening using fourth-order polynomials in order to avoid unphysical fluctuations related to the camera resolution and the contour-detection algorithm. As shown in Fig. 2c, d, the comparisons between the experimental and numerical evolutions reveal an excellent agreement. As the capillary velocity $v_{cap} = \gamma/\eta$ is independently evaluated (see ref. 32 and Supplementary Method 1 for the viscosity calibration), and typically varies between 22 and 1013 μm/s for near-critical droplets within the $\Delta T = 1$–8 K range, the precursor-film thickness $\epsilon$ remains the only fit parameter in this comparison, and its behavior with temperature will be discussed after.

Tanner's law[1,4] can be obtained from the combination of (i) a local power balance between capillary driving and viscous damping near the contact line; and (ii) global volume conservation. The former power balance reduces to the Cox–Voinov's law for total-wetting conditions:

$$\theta^3 = 9\ell\mathrm{Ca},\qquad(2)$$

where $\mathrm{Ca} = \dot{R}/v_{cap}$ is the capillary number, and $\ell = \ln(L/\epsilon)$ is the logarithmic factor discussed in the introduction relating the two cut-off lengths of the problem, namely a typical macroscopic size $L$ of the system and a microscopic length which is identified to the precursor-film thickness $\epsilon$ in total-wetting conditions.

To disentangle evaporation, through the $V(t)$ behavior obtained in Fig. 2a, from the spreading dynamics, we introduce the following dimensionless variables:

$$\tilde{R} = R\left[\frac{\pi}{4V(t)}\right]^{1/3}\quad,\quad \tilde{t} = v_{cap}t\left[\frac{\pi}{4V(t)}\right]^{1/3}.\qquad(3)$$

Tanner's law[1,4] is then written in dimensionless form as:

$$\tilde{R}^{10} - \tilde{R}_0^{\,10} = \frac{10}{9\ell}\tilde{t},\qquad(4)$$

with $\tilde{R}_0 = \tilde{R}(t = 0)$. In Fig. 3a, we plot the rescaled contact radius as a function of rescaled time, for various temperatures. We systematically observe Tanner behaviors. Interestingly, from the fits to Eq. (4), we obtain increasing values of $\ell$ as the critical point is neared. This trend is further confirmed in Fig. 3b, where we see Cox–Voinov behaviors (see Eq. (2)) at large-enough Ca, with an identical evolution of $\ell$ with $\Delta T$. The departure from Cox–Voinov's law at low Ca is due to the evaporation-induced non-monotonic behavior of $R(t)$ (see Fig. 2b), resulting in Ca crossing zero for finite values of $\theta$.

Using the $\ell$ values obtained from the Tanner fits above, we can then represent all the data onto a single master curve, as shown in Fig. 4a. The observed collapse, over more than two orders of magnitude in time and broad ranges of material parameters, shows the surprising robustness of Tanner's law in the vicinity of the critical point, despite the increasing roles of evaporation, gravity, and fluctuations.

Finally, Fig. 4b shows the extracted precursor-film thickness $\epsilon$ as a function of $\Delta T/T_c$. Strikingly, over the considered temperature range, we observe a sharp decrease of $\epsilon$ from a fraction of micrometers down to a nanometer, as the critical point is approached from above. Over a decade in the considered temperature range, this behavior is consistent with the empirical power law $\epsilon = a(\Delta T/T_c)^{2.69}$, where $a = 5.54$ mm. The wetting transition in our system being located far above the largest temperature studied here[37], we could have instead expected an increase of $\epsilon$, since the precursor film is here made of the most-wetting phase[3,39]. Nevertheless—provided that one can extrapolate the definition of interfaces towards the critical point—the spreading parameter[1] is expected to strictly vanish at that point since the two fluid phases become indistinguishable media, which might be the reason underlying

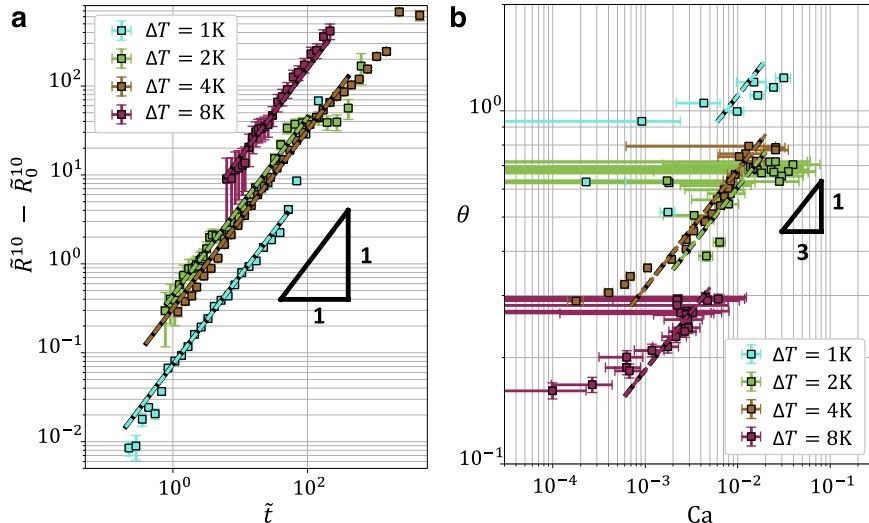

**Fig. 3 | Rescaled data: Tanner and Cox–Voinov laws. a** Rescaled contact radius $\tilde{R}^{10} - \tilde{R}_0^{\,10}$ as a function of rescaled time $\tilde{t}$, for various temperatures $\Delta T$ as indicated. The dashed lines indicate fits Eq. (4), with $\ell$ as a free parameter for each temperature. **b** Contact angle $\theta$ as a function of capillary number Ca, for various temperatures $\Delta T$ as indicated. The dashed lines indicate the predictions of Eq. (2), using the $\ell$ values obtained from the fits in **a**. Error bars on rescaled radius and contact angle are obtained using the errors on droplet height and contact radius described in Supplementary Method 2. Source data are provided as a Source Data file.

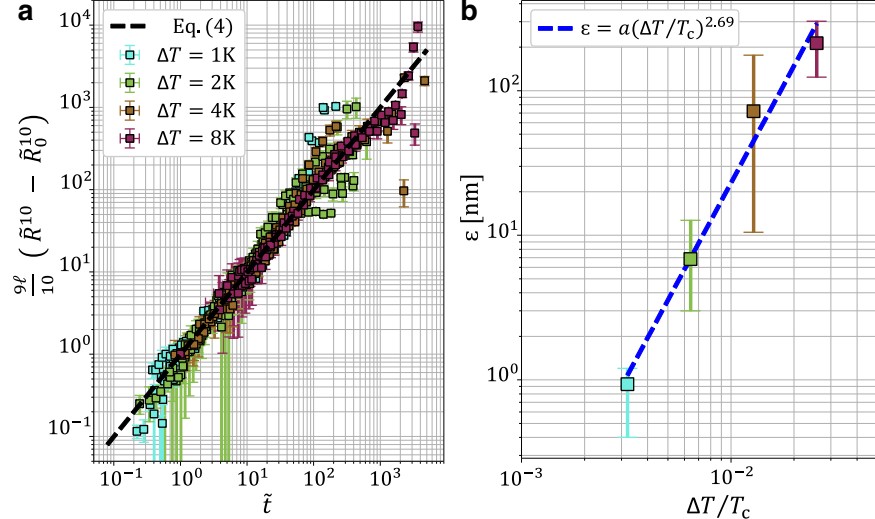

**Fig. 4 | Near-critical Tanner masterplot and precursor-film thickness.**
**a** Dimensionless Tanner master curve, including the experiments at all temperatures. The dashed line corresponds to Eq. (4). The values of the logarithmic factor $\ell$ were first obtained by fitting the individual experimental data in Fig. 3**a** to Eq. (4). Error bars on rescaled radius are obtained using the errors on droplet height (through volume derivation) and contact radius described in Supplementary Method 2.

**b** Extracted precursor-film thickness $\epsilon$ as a function of the temperature distance $\Delta T$ to the critical point, as obtained by fitting individual experimental profiles to numerical solutions of Eq. (1) (see Fig. 2**c** and **d**). The dashed line indicates the empirical power law $\epsilon = a(\Delta T/T_c)^{2.69}$ with $a = 5.54$ mm. Error bars on precursor-film thickness correspond to the maximum acceptable values to fit numerical profiles with the experimental ones. Source data are provided as a Source Data file.

the observed behavior. Beyond revealing the universality and peculiarities of viscous spreading near a critical point, our results pave the way toward further investigations closer to that point, with the aim of addressing the increasing contributions of gravity, evaporation, and eventually thermal fluctuations[40–42].

## Methods

Supplementary Method 1 presents the near-critical properties of the binary liquid used and Supplementary Method 2 the technical details for the edge detection of interfaces and data acquisition from image analysis.

## Data availability

Source data are provided with this article. Source data are provided with this paper.

## Code availability

Information on the numerical codes can be provided by the authors upon request.

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

## Acknowledgements

The authors thank Julie Jagielka, Eloi Descamps, Thomas Guérin, and Yacine Amarouchene, for preliminary work and interesting discussions, as well as the LOMA mechanical and electronic workshops for their technical contributions. The authors acknowledge financial support from the European Union through the European Research Council under EMetBrown (ERC-CoG-101039103) grant to T.S. Views and opinions expressed are however those of the authors only and do not necessarily reflect those of the European Union or the European Research Council. Neither the European Union nor the granting authority can be held responsible for them. The authors also acknowledge financial support from the Agence Nationale de la Recherche under FISICS (ANR-15-CE30-0015-01) grant to J.-P.D. and U.D., and EMetBrown (ANR-21-ERCC-0010-01), Softer (ANR-21-CE06-0029), and Fricolas (ANR-21-CE06-0039) grants to T.S. Finally, they thank the Soft Matter Collaborative Research Unit, Frontier Research Center for Advanced Material and Life Science, Faculty of Advanced Life Science at Hokkaido University, Sapporo, Japan.

## Author contributions

R.S., T.S., and J.-P.D. conceived the research plan. R.S. and A.B. performed the experiments, C.P. performed the simulations, and A.C. and U.D. participated in the data analysis with all the authors. The manuscript was written with the contributions of all the authors.

## Competing interests

The authors declare no competing interests.
