## [Peer Review File · Nature Communications]

REVIEWER COMMENTS

Reviewer #1 (Remarks to the Author):

Saiseau et al. performed an experimental investigation on the spreading of compound microdrops on solid surfaces immersing in the liquid phase. The key finding is that the classical Tanner's law, which describes the spreading dynamics of Newtonian fluids in the viscous regime, is also valid for complex liquid mixtures near their phase separation. By fitting the experimental results, the authors also extrapolated the cut-off length in the Tanner's law, which has been considered as the precursor film thickness and was found to decrease sharply when approaching the critical point. This work is interesting but the topic is too specific. Detailed comments are listed below.

1. The experimental setup is quite ingenious, but also complicated. It brings a lot of undetermined/unknown experimental parameters, including the composition of the spreading droplet, and its surface tension and viscosity.
2. The spreading process is accompanied with phase separation, surfactant reorganization, and liquid dissolution, which is very complex. Though a seemingly good agreement between the experimental and theoretical results is reached, it is hard to witness the realistic spreading process.
3. On page 3, the authors observed a Tanner-like power law for spreading and thus claimed that the evaporation is quasi-static. This statement does not make sense as it can be clearly seen that the droplet volume dramatically decreases through the entire process (inset of Fig. 2A), particularly in the early stages, where the power-law behavior is observed.
4. How does the surfactant influence the spreading process? Any effects of the chemical potential on the spreading process, which is accompanied with dissolution?
5. What exactly the liquid viscosity and surface tension are?
6. The viscous dissipation of the surrounding liquid is neglected in their modeling. Are there some evidences?
7. The linear correction between the contact angle and the capillary number in Fig. 3b is not as good as expected from the original format of the Tanner's law (i.e. eq. 2), particularly for the case close to the critical temperature (see $\Delta T=1$ K in Fig. 3b), which suggests that some other effects cannot be ignored as did by the authors.
8. Obviously, the experimental data at $\Delta T=1$ K in Fig. 3b cannot be described by the Tanner's law at all, but the extracted precursor film thickness owns a very small error. Why?

In conclusion, this manuscript is well organized and written, but the underlying physics of the spreading is not sufficiently convincing, as the phenomenon is rather complex.

Reviewer #2 (Remarks to the Author):

The experimental and numerical work presented in this manuscript deal with spreading of oil-water picoliters droplets near critical conditions. Droplets interfaces are assumed to be spherical caps and are followed during the spreading at different temperatures near the critical point. Experimental results are in excellent agreement with the numerical modelling.

The reviewer would like the authors to check their error bars on Fig 4B. Indeed the errors bars for the purple (1 nm) and blue (10 nm) points are almost invisible while they are clearly visible for the yellow and red points above 30 nm.

Reviewer #3 (Remarks to the Author):

In their manuscript, the authors experimentally investigate the spreading of droplets created in a near-critical mixture on a substrate. The results are compared by the theoretical prediction, i.e. Tanner's law, showing remarkable agreement over several orders of magnitude. The experimental droplet generation method is sophisticated and all the general approach and methods, the data acquisition and the evaluation are valid, the discussion and comparison with theoretical approaches is sound.

Due to the neat agreement with Tanner's law and the scarcely investigated liquid-in-liquid configuration of spreading, the manuscript is an appealing and novel contribution to the community.

The reviewer has only a few minor remarks:

1) The authors introduce a Heaviside step function and a not well specified function $f(t)$ into their lubrication approach in Eq. (1). The reviewer wonders why the Heaviside function is required at all. Why should the droplet only lose mass if the non-dimensional radius is less than 1?

2) Also, is the function $f(t)$ just based on the single fit $(1-t/t_f)^{1.77}$ of Fig. 2A? Or is a new fit done for each experiment, i.e. different $f(t)$ for different (ΔT). Can the authors discuss in more detail how this mass loss can be calculated based on a theoretical model? The scaling is well captured by comparison with Ref. [11], but it would be helpful if one could estimate also the amount of mass transfer somehow, not only the scaling.

3) The authors attribute the mass loss of the droplet to "evaporation". However, the mass is not lost to a gas phase, as "eVAPORation" suggests. Would not "dissolution" or just "mass transfer" be a more accurate wording? Otherwise, the reader might be misled that there is indeed a gas phase present.

4) Having Fig. 1D(b) in mind, the reviewer wonders how strong the results depend on the parameters for e.g. the Canny filter. In particular, the following plots (Fig 2 & 3) have very small error bars. Are the authors indeed sure that the error bars reflect the uncertainty by the image analysis well?

– Referee response –

R. Saiseau et al.

We would like to thank all the Referees for their work, and we hope that our response, together with the revised manuscript and an additional SI document, are convincing and that the article is found suitable for publication in Nature Communications.

REFeree 1

Saiseau et al. performed an experimental investigation on the spreading of compound microdrops on solid surfaces immersing in the liquid phase. The key finding is that the classical Tanner's law, which describes the spreading dynamics of Newtonian fluids in the viscous regime, is also valid for complex liquid mixtures near their phase separation. By fitting the experimental results, the authors also extrapolated the cut-off length in the Tanner's law, which has been considered as the precursor film thickness and was found to decrease sharply when approaching the critical point. This work is interesting but the topic is too specific. Detailed comments are listed below.

We thank the Referee for their work and careful analysis of our article. We are pleased to read that the Referee finds the work interesting and appreciates our original finding related to the validity of Tanner's law near a critical point. The Referee however raised one main concern about the specificity of the topic, which we address hereafter, before answering the more detailed questions.

Reading the detailed comments of the Referee, we suppose that the "specificity" mentioned by the Referee is related to our chosen experimental system. The latter may indeed seem specific at first sight, but we argue hereafter that this not the case. An important thermodynamic fact is that, when working close to criticality, one can choose *any* fluid belonging to the ($d=3$, $n=1$) universality class of the Ising model and generalize results to *all* fluids of the same class. Therefore, our investigations using micellar phases of microemulsions – despite apparently specific – is in contrast fully generalizable to any isotropic liquid when approaching the critical point. As an important consequence, our study reveals and quantifies for the first time the universal spreading dynamics of droplets of any fluid of the mentioned universality class. Strikingly, we robustly reveal that the celebrated classical Tanner's law of ideal droplet spreading remains valid very near criticality, despite the complexity associated with "evaporation" (N.B.: this terminology is discussed below), gravity and thermal fluctuations. Moreover, we also analyze the universal behavior of the precursor film present

ahead of the droplets and quantitatively demonstrate a counter-intuitive abrupt vanishing of the latter at criticality. As such, our study is in fact universal and reveals a novel fundamental physical effect by the vanishing of equilibrium prewetting films when approaching the critical point. We therefore would like to maintain our view that the results are universal and as such would appeal to a broad readership.

Nevertheless, we agree with the Referee that the universality of our study was not sufficiently clear in the manuscript, which could be a reason for the confusion on specificity vs universality. To address this, we have clarified the manuscript and included a new, complete Supplementary Information (SI) document, where the first section is devoted to “near-critical micellar phases of microemulsion”. Universality close to a critical point is described, as well as the properties of particular interest for near-critical droplet spreading in the two-phase region, along with key references.

We hope the Referee is satisfied with our detailed answer and edits, and now understands in a better way our significance statement. Below, we address the other, more detailed comments.

1. The experimental setup is quite ingenious, but also complicated. It brings a lot of undetermined/unknown experimental parameters, including the composition of the spreading droplet, and its surface tension and viscosity.

We are glad to read that the Referee finds our experimental setup “ingenious”. Below, we address the concern on complexity, as well as the fair remark on the missing experimental parameters.

As explained above, the main reason which brought us to use isotropic critical fluids is the emergence of universal behaviors close to the critical point. Thus, beneath the apparent complexity and specificity, the system is in fact universal in its behaviours. The choice of a binary liquid, instead of a liquid-gas system, relates to easier technical manageability, in particular in temperature control. The choice of a micellar phase of microemulsion as a binary liquid relates to the fact that the typical microscopic length scale ξ_0 is typically ten time larger in supramolecular liquids than in classical molecular ones (nanometric micelles versus angström-sized molecules) leading to a strong decrease of the amplitude γ_0 of the interfacial tension. Indeed, an established critical scaling expression shows that $\gamma_0 \xi_0^2 / k_B T$ is a universal constant [Modolva, *Phys. Rev. A* 31, 1022 (1985)]. Finally, although microemulsions are multicomponent systems (including water, toluene, SDS and butanol), when water is only 9% wt and SDS is only 4% wt, as compared to 87% wt for toluene + butanol, not only micelles are formed to give birth to a binary-like mixture, but also this mixture presents a critical line belonging to the (d=3, n=1) universality class of the Ising model.

As far as the experimental parameters are concerned, this microemulsion has been well characterized in previous publications, that are cited in the manuscript. In addition, we have characterized the refractive index contrast and the susceptibility, despite that these are not central to the present investigation. Below, and in the SI document, we provide all the requested details.

(i) Composition: to perform an experiment, the system is first set at a thermodynamic equilibrium at a temperature T above the critical one T_c , in a phase-separated state. So, as given in the SI, the concentrations of the two micellar phases are perfectly known:

$\Phi_{i=1,2} = \Phi_c + b \left(\frac{T-T_c}{T_c} \right) \pm \frac{\Delta\Phi_0}{2} \left(\frac{T-T_c}{T_c} \right)^\beta$, with the universal exponent $\beta = 0.325$, the concentration at criticality $\Phi_c = 0.11$, the asymmetry parameter $b = 1.185$ and the coexistence amplitude $\Delta\Phi_0 = 0.275$. Moreover, as the initial manipulation consists in using the optical radiation pressure to bring a droplet of concentration Φ_2 towards the bottom of the phase Φ_1 , then the composition of the spreading droplet is given by Φ_2 .

(ii) Interfacial tension: at a given temperature T above the critical one T_c , as given in the SI, the interfacial tension between the two phases is: $\gamma = \gamma_0 \left(\frac{T-T_c}{T_c} \right)^{2\nu}$, with the universal exponent $\nu = 0.63$ and the amplitude $\gamma_0 = (4.6 \pm 0.4) \cdot 10^{-5}$ N/m.

(iii) Viscosity: the viscosities of the two phases have been estimated at different distances in temperature to T_c . Results are the following: $\eta_{i=1,2} = [1.46 - 0.014(T - 273)] \times (1 + 2.5\Phi_{i=1,2}) \cdot 10^{-3}$ Pa. s. Besides, we added several references in the SI to explain how these quantities were robustly obtained.

2. The spreading process is accompanied with phase separation, surfactant reorganization, and liquid dissolution, which is very complex. Though a seemly good agreement between the experimental and theoretical results is reached, it is hard to witness the realistic spreading process.

We thank the Referee for this question, which is addressed below. We divide it into four sections focusing on: phase separation, surfactant reorganization, liquid dissolution and realistic process.

(i) Phase separation: the spreading is in fact not accompanied by any phase separation. Indeed, the global two-phase thermodynamic equilibrium of the two media is achieved from scratch, which is actually a key ingredient to properly address physics close to the critical point.

(ii) Surfactant reorganization: generally speaking, as the surfactant is in minority, and even of concentration smaller than that of water (respectively 4% wt SDS and 9% wt water), all the surfactant is used to form micelles. At $T > T_c$, the binary mixture separates in two coexisting micellar phases, one rich in micelles with concentration Φ_1 , and the other poor in micelles with concentration Φ_2 . This is classical to binary mixtures, which show the coexistence of two phases, one rich and the other poor in solute. In our case, the solute is represented by the micelles. Consequently, surfactant does not reorganize during “liquid dissolution” (see point (iii) below). Instead, this is the concentration in micelles that locally varies. Moreover, as phase separation is dictated by equality between the chemical potentials of the two phases, the interface does not present any extra surfactant, since all are involved in the micelles, and surfactants thus do not contribute directly to the interfacial tension.

(iii) Liquid dissolution: on the one hand, the system is first set at a thermodynamic equilibrium at $T > T_c$, in a phase-separated state, so the critical radius for the nucleation of droplets constituted by one phase into the other is infinite, as asymptotically demonstrated by the Ostwald-ripening theory (evaporation of droplets with radii smaller than the evolving critical radius, at the expense of larger ones). On the other hand, using the radiation pressure as a tool, we handle a finite volume drop of phase Φ_2 , in the large volume of phase Φ_1 . Consequently, this drop cannot be stable and must evaporate because there is an extra term in its chemical potential related to curvature (see the classical Gibbs-Thomson relation) which prevents chemical-potential equality of the two phases. Therefore, “evaporation” or “liquid dissolution” (see dedicated discussion on the terminology below) is necessary to recover equilibrium. This is just a classical thermodynamic effect related to mass-conserved systems. There is no particular conceptual issue associated with it.

(iv) Realistic process: We now finally turn to the comment by the Referee on the realistic spreading process. We measure an enlargement of the drop contact radius in time, with (i) no phase separation, (ii) no effect of surfactant, and (iii) a measured and precisely quantified evaporation. Moreover, by renormalizing the data in order to remove the effect of the evaporation, we robustly recover the classical Tanner’s law of spreading and also obtain a perfect agreement with theoretical profiles from the minimal lubrication theory. We can therefore safely conclude that the law of spreading is well characterized here by Tanner’s law without any artefact.

As a final remark, we recall that there was a concluding discussion paragraph in the manuscript with possible non-exclusive scenarios for the vanishing of the precursor film. While the spreading-parameter scenario is certainly correct, the Casimir one was – admittedly – only a tentative proposition for discussion. However, since the Referee is here concerned with the realistic mechanism at play in the study, we have decided to remove the mention of this hypothetical scenario. As such, everything presented now is self-consistent and realistic with no major hypothesis needed.

3. On page 3, the authors observed a Tanner-like power law for spreading and thus claimed that the evaporation is quasi-static. This statement does not make sense as it can be clearly seen that the droplet volume dramatically decreases through the entire process (inset of Fig. 2A), particularly in the early stages, where the power-law behavior is observed.

We thank the Referee for this remark. We agree that our previous terminology was misleading. What we wanted to express by that is the following. Tanner’s law states that, asymptotically, at long times, one has: $R(t) \sim \left(\frac{V^3 \gamma t}{\eta}\right)^{1/10}$. In our study, we demonstrate that this law is still valid for critical binary mixtures, provided that we simply substitute the constant droplet volume V in Tanner’s law by the evaporating one $V(t)$. As such, we have an “adiabatic” evaporation within the spreading law. In other words, evaporation is not fast enough to affect the spreading scaling exponent on several decades. This is what we meant by “quasi-static”.

Furthermore, evaporation is controlled by the diffusion of the first phase into the second one. A direct consequence of the assumption of steady diffusion-driven evaporation for small drops is that the rate of change of volume is proportional to the drop radius, *i.e.* $dV/dt = -2\pi J_0 \cdot R$, where J_0 is the evaporation parameter, or the diffusion coefficient (see *Cazabat and Guena, Soft Matter 6, 2591 (2010)*). This relation can be checked experimentally by integrating both sides of the equality versus time, as shown in Fig. R1.

Figure R1: Drop volume $V(\tau)$ as a function of the time integral of its contact radius $R(\tau)$, where $\tau = t_f - t$, for a spreading-drop experiment performed at $T - T_c = 8$ K. The solid red line is a linear fit (*i.e.* a line with slope 1 in logarithmic representation).

Figure R1 clearly shows that the volume $V(t)$ is indeed proportional to the time integral of the contact radius $R(t)$. This behavior was further checked to be valid for all the temperatures investigated in our study. Hence, the evaporation process can be truly described by a steady diffusion. Moreover, the latter is slow enough (characteristic time $\tau \sim 200$ s) to ensure a quasi-static spreading of the droplet. Naturally, evaporation will start to be inevitable at some point. That is why the contact line finally recedes, as described in [*Cazabat and Guena, Soft Matter 6, 2591 (2010)*]. As mentioned in the manuscript, we have experimentally characterized the variation of the drop volume all along the evaporation process, and we found $V(t) = V_0 \cdot (1 - t/t_f)^{1.77}$, the exponent of which is very close to the exponent 11/7 theoretically obtained in [*Cazabat and Guena, Soft Matter 6, 2591 (2010)*] for a stationary evaporation process.

All together, we can safely describe our experimental results through viscocapillary flow, and in particular from Tanner's law. In addition to that, we monitored and fully rationalized the

evaporation process, and showed that it is simply resulting from the steady diffusion process expected from pure and established equilibrium arguments [Cazabat and Guena, *Soft Matter* 6, 2591 (2010); Jambon-Puillet et al., *Journal of Fluid Mechanics* 844, 817 (2018)]. However, we agree with the Referee that our sentence on quasi-steadiness was perhaps misleading. Moreover, it is not needed for our demonstration and we have thus removed it from the revised manuscript.

4. How does the surfactant influence the spreading process? Any effects of the chemical potential on the spreading process, which is accompanied with dissolution?

Answers to these questions are addressed separately below.

(i) Surfactant influence on the spreading process: as explained above, surfactants are sufficiently diluted and only important to create and thermodynamically stabilize micelles that constitute the “solute” in the two coexisting phases of concentrations Φ_1 and Φ_2 . Consequently, there is no “free” surfactant available to adsorb at any surface or interface.

(ii) Effects of the chemical potential on the spreading process, which is accompanied with dissolution: as discussed above, we necessarily have a local imbalance of chemical potentials. Indeed, a small volume, *i.e.* the one of the spreading drop, is extracted from of a large reservoir of one phase by the external work of radiation pressure, and is further set in contact with another reservoir constituted by the other phase. Moreover, the two reservoirs are in equilibrium with identical chemical potentials. Hence, the droplet’s chemical potential is different (higher) than the one of its initial reservoir. This imbalance is well described by the classical Gibbs-Thomson relation. It also directly explains why the spreading drop inevitably evaporates, since the equality of chemical potentials must be recovered to reach thermodynamic equilibrium at the end. So, we fully agree with the Referee that there naturally exists an effect of the local chemical potential, leading *e.g.* to evaporation. However, the spreading itself is unaffected by that, nor by surfactants since the latter do not play any role on the surface tension here (see previous point (i)).

5. What exactly the liquid viscosity and surface tension are?

We apologize for the missing information, which we agree to be useful and important to the readers. As discussed above, and provided in the additional SI document, we have:

(i) Viscosity: $\eta_{i=1,2} = [1.46 - 0.014(T - 273)] \times (1 + 2.5\Phi_{i=1,2}) \cdot 10^{-3}$ Pa.s ;

(ii) Interfacial tension: $\gamma = \gamma_0 \left(\frac{T-T_c}{T_c}\right)^{2\nu}$ with the universal exponent $\nu = 0.63$ and the amplitude $\gamma_0 = (4.6 \pm 0.4) \cdot 10^{-5}$ N/m.

6. The viscous dissipation of the surrounding liquid is neglected in them modeling. Are there some evidences?

We thank the Referee for this fair question. It is important to realize here that, in the theory of spreading, viscous dissipation is neglected mostly everywhere in fact, *i.e.* outside the

droplet but also in the bulk of the latter too. This is due to the following important aspect. Tanner's law is an asymptotic expression, valid for total wetting conditions, at late times, and for vanishing dynamical contact angles. In such a situation, the dissipation is mostly located in the wedge-like region of the droplet near the contact line. Indeed, as realized early by Huh and Scriven in their famous recognition of the "wetting paradox" related to the no-slip condition at the substrate [C. Huh and L. E. Scriven, *Journal of colloid and interface science* 35, 85 (1971)], the typical viscous stress scales as $\sim \eta V/h$ where h is the vanishing fluid gap in the wedge. As such, the dissipation is expected to diverge at the contact line, within the droplet. Of course, this infinity associated with continuum mechanics, eventually breaks down and a molecular length scale is needed to regularize that expression – as explained in the manuscript. Nevertheless, the continuum argument still provides the main picture beyond logarithmic corrections: the viscous dissipation is dominated by the wedge flow. This scenario is mainly not modified in the presence of an atmospheric fluid of similar viscosity. We however stress that the picture would be modified if the outer viscosity was to be much bigger than the droplet one. Fortunately, this is not the case near the critical point, where both viscosities are almost equal. We have added such an explanation in the revised manuscript.

7. The linear correction between the contact angle and the capillary number in Fig. 3b is not as good as expected from the original format of the Tanner's law (i.e. eq. 2), particularly for the case close to the critical temperature (see $\Delta T=1$ K in Fig. 3b), which suggests that some other effects cannot be ignored as did by the authors.

The Referee raises here an important experimental difficulty we had to face: as compared to the robust global measurements on volume and contact radius, local contact-angle measurements are extremely challenging when getting closer and closer to the critical point. We discuss this in detail in the following paragraphs, and then conclude on the potential need for additional effects.

(i) Optical resolution is limited by the fact that we need to image the spreading drop with long working distance objectives, because the sample is inserted in a thermo-regulated oven in brass – which has some minimum thickness. Therefore, the numerical aperture is small and resolution is limited.

(ii) Even more important: we lose more and more contrast when the critical point is approached for two reasons. On the one hand, the refractive-index contrast between the two phases is low, it decreases, and it even vanishes at T_c . Specifically, it behaves as $n_2 - n_1 = 0.045 \left(\frac{T-T_c}{T_c}\right)^\beta$, with the universal critical exponent $\beta = 0.325$. On the other hand, the correlation length $\xi = \xi_0 \left(\frac{T-T_c}{T_c}\right)^{-\nu}$ of density fluctuations, and the susceptibility $\chi_T = \chi_0 \left(\frac{T-T_c}{T_c}\right)^{-\gamma}$, with $\chi_0 = 1.344 \cdot 10^{-6} \text{ Pa}^{-1}$ and $\gamma = 1.24$, both increase and even diverge when the critical point is neared. Thus, the light scattering losses represented by the turbidity (proportional to $\chi_T f(\xi)$ where f is a function) of the phases increases and diverges too (see Ref. [33]). This mechanism blurs the imaging when decreasing $(T - T_c)$, much faster than the index contrast vanishes (critical exponent 1.24 versus 0.325). Consequently, while contact angles are never easily measured, the use of near-critical mixtures tremendously increases

such a difficulty. We therefore had to infer the local contact angles from global volume and radius data, and this fact very likely explains the observed relatively reduced quality of the Cox-Voinov plot as compared to the Tanner plot. Nevertheless, we would like to stress that there is no adjustable parameter in the Cox-Voinov curves, as the logarithmic factor were fixed prior to these curves and are the exact same as the ones for the corresponding Tanner curves. Therefore, we maintain our view that, within our optical and angle resolutions, the power-law part of the angle-speed data is in good agreement with Cox-Voinov's law. We do not need other effects to rationalize the data, and such novel effects would anyway modify the Tanner plot, for which there is no doubt about its validity over several decades.

8. Obviously, the experimental data at Delta T=1 K in Fig. 3b cannot be described by the Tanners' law at all, but the extracted precursor film thickness owns a very small error. Why?

We thank the Referee about this question, and we separate it in two parts to address it properly.

First, the Referee questions the validity of Tanner's law for the 1 K data (purple symbols). We unfortunately disagree with this statement. While the long-term data is indeed dominated by evaporation – a well-understood, natural and inevitable process that was discussed above in details – the short-term data is in contrast perfectly captured by Tanner's law over at least a decade in time. Perhaps, the Referee had instead in mind the Cox-Voinov's law? If that would be the case, we already explained the difficulty in extracting angles near the critical point in our response to the previous point, thus the error source at 1 K from that point.

Secondly, we thank the Referee for the great catch about the error bars. Indeed, that was a typo on our side and we apologize for that. This comment led us to redo in depth the data analysis leading to all figures, including the error-bar analysis. Error bars have now been corrected in all the figures of the revised manuscript. The results are unchanged, apart from a slight modification of the numerical parameters of the empirical fit in Fig. 4B.

In conclusion, this manuscript is well organized and written, but the underlying physics of the spreading is not sufficiently convincing, as the phenomenon is rather complex.

We thank once again the Referee for their time, and the careful analysis of our manuscript. We appreciate the positive comment on the organization and writing qualities of the manuscript. Finally, we hope that our detailed response, the new complete SI document and the revised manuscript, altogether help providing the missing information on the critical-physics side of our work. We hope the ensemble is now convincing, and that the Referee will now share our excitement towards these results.

REFEREE 2

The experimental and numerical work presented in this manuscript deal with spreading of oil-water picoliters droplets near critical conditions. Droplets interfaces are assumed to be spherical caps and are followed during the spreading at different temperatures near the critical point. Experimental results are in excellent agreement with the numerical modelling.

We thank the Referee for their work and for the critical reading of our manuscript. We are pleased to read that the Referee finds the experimental data and numerical simulations to be in excellent agreement. The Referee has only one minor remark, that is addressed below.

The reviewer would like the authors to check their error bars on Fig 4B. Indeed the errors bars for the purple (1 nm) and blue (10 nm) points are almost invisible while they are clearly visible for the yellow and red points above 30 nm.

We thank the Referee for this great catch, that was also identified by the other Referees. This is indeed a typo on our side, which we apologize for. This comment led us to redo in depth the data analysis leading to all figures, including the error-bar analysis. Error bars have now been corrected in all the figures of the revised manuscript. The results are unchanged, apart from a slight modification of the numerical parameters of the empirical fit in Fig. 4B.

REFEREE 3

In their manuscript, the authors experimentally investigate the spreading of droplets created in a near-critical mixture on a substrate. The results are compared by the theoretical prediction, i.e. Tanner's law, showing remarkable agreement over several orders of magnitude. The experimental droplet generation method is sophisticated and all the general approach and methods, the data acquisition and the evaluation are valid, the discussion and comparison with theoretical approaches is sound. Due to the neat agreement with Tanner's law and the scarcely investigated liquid-in-liquid configuration of spreading, the manuscript is an appealing and novel contribution to the community.

The reviewer has only a few minor remarks.

We thank the Referee for their work and for the critical reading of our manuscript. We are delighted to read that the Referee highlights the remarkable experiment-theory agreement over several orders of magnitude, and considers the work appealing and a novel contribution to the field. Besides, we are also pleased to read that the methods are found to be sophisticated and the ensemble of the analysis to be sound. Finally, the four minor remarks by the Referee are addressed hereafter.

1) The authors introduce a Heaviside step function and a not well specified function $f(t)$ into their lubrication approach in Eq. (1). The reviewer wonders why the Heaviside function is required at all. Why should the droplet only lose mass if the non-dimensional radius is less than 1?

This is indeed a subtle aspect of the physics at play in the system. In fact, we spent a lot of time thinking about this question prior to the first submission of our manuscript. We would like to firmly maintain our view that the Heaviside function is the best candidate function in the modeling, as explained in the following.

There is a drastic distinction between the microscopic precursor film and the droplet. While the droplet is an out-of-equilibrium system by construction, which will spread and eventually completely evaporate (see discussion about chemical potentials above, in the response to Referee 1), the precursor film is a very different entity as far as equilibrium is concerned. Indeed, it consists of a prewetted fluid layer, already existing prior to the deposition of any droplet (we were able to observe its existence above T_c , despite this is not reported in the manuscript) and whose thickness is fixed for good by favorable van der Waals forces. As a consequence, the precursor film is at equilibrium, and the thickness is constant despite the (compensating) evaporation and condensation fluxes. In summary, we certainly do not want to artificially add evaporation to the prewetting film, hence the Heaviside function in the model is used in order to separate well the droplet and film regions.

However, we acknowledge the subtlety of this point and, thanks to the remark by the Referee, we have added a clarifying sentence in the manuscript.

2) Also, is the function $f(t)$ just based on the single fit $(1-t/t_f)^{1.77}$ of Fig. 2A? Or is a new fit done for each experiment, i.e. different $f(t)$ for different (ΔT). Can the authors discuss in more detail how this mass loss can be calculated based on a theoretical model? The scaling is well captured by comparison with Ref. [11], but it would be helpful if one could estimate also the amount of mass transfer somehow, not only the scaling.

We thank the Referee for this question, as we realize that our associated explanations were not clear enough in the previous version of the manuscript. This has been corrected in the novel version of the manuscript.

Mainly, as can be seen from the raw data, the final time t_f for droplet vanishing, is a function of drop volume and temperature. In practice, for each given temperature, a fit is performed in order to determine both t_f and the exponent. The important aspect is that the exponent is itself unique for all temperatures, and nearly identical to the one predicted for evaporating droplets, see Ref. [11]. As a consequence, the evaporation part of the process, for which we have no interest in the current study, is robustly quantified and understood.

3) The authors attribute the mass loss of the droplet to "evaporation". However, the mass is not lost to a gas phase, as "eVAPORation" suggests. Would not "dissolution" or just "mass transfer" be a more accurate wording? Otherwise, the reader might be misled that there is indeed a gas phase present.

We agree that the terminology might not necessarily be straightforward, but we have used the generic term raised for the universal dynamics of phase transitions. For instance, Ostwald ripening in condensed phases is very often described as an evaporation-condensation mechanism, while a gas phase is not necessarily involved. Specifically, therein, one has: condensation for a positive growth rate $dR/dt > 0$, when $R > R_c$, with R_c the critical radius; and evaporation for a negative growth rate when $R < R_c$. However, our universal terminology might indeed be misleading for the wetting community, since in our case the atmospheric fluid is liquid-like too. Therefore, we decided to define precisely what we mean by "evaporation" early in the revised manuscript.

4) Having Fig. 1D(b) in mind, the reviewer wonders how strong the results depend on the parameters for e.g. the Canny filter. In particular, the following plots (Fig 2 & 3) have very small error bars. Are the authors indeed sure that the error bars reflect the uncertainty by the image analysis well?

We thank the Referee for raising the issue with the error bars, which was also noted by the other Referees. This was a typo in our side, which we apologize for. This comment led us to redo in depth the data analysis leading to all figures, including the error-bar analysis. Error bars have now been corrected in all the figures of the revised manuscript. The results are

unchanged, apart from a slight modification of the numerical parameters of the empirical fit in Fig. 4B. Moreover, we decided to add the associated technical details about the Canny filter in the SI document.

Finally, in addition to the changes listed above and the additional SI document, note that we have also modified slightly the figures (label and marker sizes, label fonts) and added: i) an acknowledgment to a colleague in the novel version of the manuscript, and ii) three references on dynamical wetting.

REVIEWERS' COMMENTS

Reviewer #1 (Remarks to the Author):

The authors have properly answered the questions from the reviewers, and the manuscript is suitable for publication.

Reviewer #2 (Remarks to the Author):

Revised manuscript deserving publication.

Reviewer #3 (Remarks to the Author):

The authors have addressed all comments appropriately.